# Hospital Antibiotic Use during COVID-19 Pandemic in Italy

**DOI:** 10.3390/antibiotics12010168

**Published:** 2023-01-13

**Authors:** Alessandro Perrella, Filomena Fortinguerra, Andrea Pierantozzi, Nicolina Capoluongo, Novella Carannante, Andrea Lo Vecchio, Francesca Futura Bernardi, Francesco Trotta, Agnese Cangini

**Affiliations:** 1I Division Emerging Infectious Disease and High Contagiousness, D. Cotugno Hospital, 80131 Naples, Italy; 2Italian Medicines Agency (AIFA), 00187 Rome, Italy; 3Pediatric Unit, AOU Federico II Medical School University, 80131 Naples, Italy; 4Pharmacy Unit, Vanvitelli University, 80131 Naples, Italy

**Keywords:** anti-bacterial agents, antibiotic resistance, drug consumption, inpatients, COVID-19, SARS-CoV-2

## Abstract

Antimicrobial resistance (AMR) represents a major issue in healthcare being correlated to global inappropriate use of antibiotics. The aim of this study was to compare the data on hospital antibiotic consumption in 2020–2021 with those related to 2019 in order to evaluate the impact of the COVID-19 pandemic on antibiotic prescriptions and appropriate use at national level and in the different geographical areas. To estimate the consumption of antibiotics, two indicators were calculated: “number of DDD per 1000 inhabitants per day” and “number of DDD per 100 hospitalisation days”. Consumption data on antibiotics dispensed in public health facilities were based on the Italian “traceability of medicines” information flow. Data on hospitalisation days were extracted from the Italian “hospital discharge form” flow. Pearson correlation analysis was performed between the number of patients hospitalised for COVID-19 and the consumption of antibiotics in public healthcare facilities. During 2020, about 1.7 DDD/1000 inhabitants per day (12.3% of the overall consumption of reimbursed antibiotics) were dispensed exclusively in Italian hospitals (+0.8% compared to 2019). Considering the number of DDD per 100 hospitalisation days, consumption increased by 19.3% in 2020 compared to 2019. Comparing the first semester of 2020 and 2019, a decrease of DDD/1000 inhabitants per day was observed (−1.6%) at national level, with opposite trends in the different geographical areas; an increase in the use of azithromycin and carbapenems was also observed, with a stable consumption of third-generation cephalosporins. The use of antibiotics in the second semester of 2020 compared to the same period of 2019 showed a clear reduction at national level (−8.5%), appreciable to a similar extent in all geographic areas. In the first semester of 2021 compared to the same period of 2020, there was a huge reduction (−31.4%) in consumption at national level. However, the variations were heterogeneous between different geographical areas. To our knowledge, this study represents the most comprehensive analysis performed on antibiotic consumption data in hospital settings in Italy during the COVID-19 pandemic to date. Despite international and national guideline recommendations, a substantial overall increase in antibiotic prescriptions was observed during the COVID-19 pandemic, with variability in terms of geographical distribution and prescription strategies. These findings may be related to the dichotomy between perceived and real significance of guidelines, expert panels, or consensus. Therefore, new approaches or strategies to antimicrobial stewardship should be proposed.

## 1. Introduction

Antimicrobial resistance (AMR) is a serious public health problem correlated to global non-optimal use of antibiotics, both in humans and animals. Regarding the use of antibiotics in human settings, recent literature evidence suggested that the COVID-19 pandemic worsened the over-prescription and inappropriateness of antibiotic prescription, contributing to the spread of resistant bacteria worldwide [1,2,3,4,5].

According to the annual reports published by the European Surveillance of Antimicrobial Consumption Network (ESAC-Net) coordinated by the European Center for Disease Prevention and Control (ECDC), the highest antibiotic consumption in humans was recorded in the southern areas of the continent [6,7]. Particularly, based on data from 29 participating European countries in 2020, the mean total consumption (community and hospital setting combined) of antibiotics for systemic use (ATC group J01) was 16.4 DDD per 1000 inhabitants per day (country range: 8.5–28.9). Despite a statistically significant overall decrease observed during the period 2011–2020 for the mean consumption in European/European Economic Area countries (EU/EEA), a considerably larger decrease was found during 2020 (−17.6%) compared to the previous years. This change could be mainly attributable to anti-COVID-19 measures and their impact on the frequency of common bacterial infections, as well as viral infections, the latter improperly treated with antibiotics. 

At local level, the majority of the European countries reported a substantial decrease between 2019 and 2020, likely due to the pandemic, with a more evident reduction of consumption in the community (−18.3%) than in the hospital setting (−4.5%). 

In 2020, Italy was among the seven countries (together with Estonia, Greece, Hungary, Latvia, Malta, and Portugal) reporting a decrease in the community, but an increase in the hospital setting, showing an overall mean antibiotic consumption (18.4 DDD per 1000 inhabitants per day) still above the average of European countries. 

Italy was one of the first Western countries greatly affected by the pandemic, particularly the northern regions where the pandemic first started. Despite the COVID-19 pandemic putting tremendous pressure on the capacity of the national hospital systems, with a rapidly increasing demand for intensive care units [8,9,10] in many hospitals, the number of patients admitted for elective surgery or chronic disease decreased during the pandemic [11,12].

Although international evidence-based guidelines [13,14,15,16] have discouraged and rationalized the use of antibiotics in patients with mild, moderate, or severe COVID-19 without suspicion of bacterial co-infection, the use of antibiotics continues to be substantial. Some preliminary findings focusing on co-infections in COVID-19 patients showed that the bacteria most frequently isolated from induced sputum and/or bronchoalveolar lavage were *Mycoplasma pneumoniae*, *Pseudomonas aeruginosa*, and *Haemophilus influenza* [17]. However, some studies (mainly from non-EU/EEA countries) reported an antibiotic prescription in COVID-19 patients significantly higher than the prevalence of bacterial co-infection, suggesting over-prescribing in some settings [18,19].

According to this contrasting evidence in the literature, showing a different approach to the use of antibiotics in clinical practice and in emergency, it seems clear how obtaining a well-defined trend of antibiotic consumption during the pandemic may help to evaluate the implications for AMR. However, a report on a large cohort of patients on the above-mentioned issue has not yet been proposed. In Italy, the Medicines Utilisation Monitoring Center (Osservatorio Nazionale sull’impiego dei Medicinali, OsMed) of the Italian Medicines Agency (Agenzia Italiana del Farmaco, AIFA) promotes the optimal use of antibiotics for human use, monitoring their national consumption and appropriate use, in line with the 2017–2020 National Action Plan on Antimicrobial Resistance (Piano Nazionale per il Contrasto dell’Antimicrobico-Resistenza, PNCAR) and the global World Health Organisation (WHO) Action Plan [20]. In this context, a national report on antibiotics use in humans is published annually by AIFA [21,22]. The National Plan defined a set of indicators of appropriateness to be monitored in hospital settings as a reduction of at least 5% of antibiotics use. 

The aim of the present study is to compare 2020–2021 national antibiotic consumption in Italy and across geographical areas with 2019 data, mainly focusing on hospital settings where the impact of the COVID-19 pandemic and the spread of multi-resistant bacteria may be more relevant.

## 2. Results

The overall consumption of antibiotics reimbursed by National Healthcare Service (NHCS) and dispensed both in community and hospital settings was 13.8 DDD/1000 inhabitants per day, showing a remarkable decrease (−21.7%) compared to 2019 (Table 1). An increasing geographical gradient of northern–southern was observed in the number of DDD/1000 inhabitants per day (South: 17.0; Center: 14.2; North: 11.4).

Overall, 12.3% (corresponding to 1.7 DDD/1000 inhabitants per day) of the total consumption reimbursed by the NHCS was dispensed exclusively in hospital settings, remaining stable at national level (+0.8%), although with an opposite trend among geographical areas (North +7.3%, Center −8.1%, and South −4.5%). Conversely, antibiotic consumption, expressed as DDD per 100 hospitalisation days, increased (+19.3%) in 2020 compared to 2019. This increase was greater in the northern (+24.7%) and the southern areas of Italy (+19.9%) than in the Center (+4.3). 

Particularly, this trend was found during a time period characterized by reduced hospital days; in fact, during 2020 in Italy, about 35.5 million hospital days were provided by public health facilities, while in 2019, the hospital days were 42.0 million. 

Comparing the first semester of 2020 and 2019, a decrease of DDD/1000 inhabitants per day was observed (−1.6%) at national level, with opposite trends in the different geographical areas; in the North, there was an increase of 5.5%, mainly determined by the changes in Lombardy (+13.3%) and Emilia-Romagna (+21.9%), while in the Center and in southern Italy, a reduction of −10.5% and −9.7% was observed, respectively (Table 2). The use of antibiotics in the second semester of 2020 compared to the same period of 2019 showed a clear reduction at national level (−8.5%), appreciable to a similar extent in all geographic areas. In the first semester of 2021 compared to the same period of 2020, there was a huge reduction (−31.4%) in the consumption of antibiotics calculated on the resident population (DDD/1000 inhabitants per day), with relevant differences by geographical area: −38.0% in the North, −23.6% in the Center, and −20.6% in the South. 

Finally, by evaluating consumption on a monthly basis, there was a peak in March 2020, with a double value compared to the same month of 2019 (4.0 vs. 2.0 DDD/1000 inhabitants per day, Figure 1). This trend was explained by the need of hospitals to quickly acquire large stocks of antibiotics to address the onset of the pandemic emergency. In the following months, consumption fell rapidly with a negative peak in May 2020 compared to the same month of the previous year (0.7 vs. 1.9 DDD/1000 inhabitants per day), due also to the evidence-based clinical recommendations provided by AIFA against the use of antibiotics for the treatment of COVID-19 patients, except for specific clinical conditions such as bacterial co-infections. From August 2020, consumption was comparable to that registered in the previous year. In the first semester of 2021, consumption was systematically lower than in 2019, with a difference that tended to decrease until it reached similar values observed in June (1.5 vs. 1.8 DDD/1000 inhabitants per day). 

### 2.1. Azithromycin

In the first semester of 2020, compared to the same period of the previous year, a relevant increase in the use of azithromycin was observed, higher in the North (+192.0%) and in the South (+145.6%) than in the Center (+69.1%) (Table 3). In the second semester of 2020, the consumption of azithromycin recorded a further increase compared to that observed during 2019 (+71.5%), although it was smaller than that observed during the first semester of the year. The variations were heterogeneous between geographical areas: the South had a higher increase (+221.2%) compared to the Center (+57.4%) and the North (+30.2%).

In the first semester of 2021, the consumption of azithromycin decreased compared to same period of 2020. The most remarkable reductions were observed in the North (−83.7%), compared to the Center (−67.7%) and the South (−63.7%). 

Analysing the consumption of azithromycin on a monthly basis, a peak was observed in the months of March and April in 2020; levels then returned to those of the period May-September 2019, and subsequently, experienced a growth with peaks in October and November (Figure 2). In the period January–August 2021, consumption was constantly lower than that recorded in 2019.

### 2.2. Third-Generation Cephalosporins

Similar to what was observed for the overall antibiotic category, consumption of third-generation cephalosporins in the first semester of 2020 was stable at national level (+0.5%) compared to that of the same period of the previous year; however, a relevant difference was observed among the three geographical areas (Table 3): consumption increased in the North (+20.9%), particularly in Lombardy (+29.5%) and Emilia-Romagna (+75.3%), while in the Center and in the South, there was a reduction by 13.3% and 21.8%, respectively. Comparing the second semester of 2020 with the same period of 2019, a similar trend to that observed in the first semester was found, even with smaller variations. On the other hand, the first semester of 2021 was characterized by reductions in all geographical areas compared to the same period in 2020, even if they were more evident in northern Italy. 

When evaluating antibiotic consumption in relation to hospitalisation days, a remarkable increase in 2020 in comparison to 2019 was found, with a greater increase in the North (+41.7%) than in the South (+9.8%) and in the Center (+2.8%). This increase could be partially explained by overuse of third-generation cephalosporins as a prophylactic approach.

### 2.3. Carbapenems

Regarding carbapenems, in the first semester of 2020 there was an increase in consumption in all geographical areas, more evident in the North (+32.8%) and in the south (+25.3%) compared to the Center (+19.0%) (Table 3). In the second semester of 2020, consumption further increased in the North (+33.5%), while remaining stable in the Center and decreasing in the South (−25.5%). In the first semester of 2021, consumption was characterized by a slight growth (+3.1%) at national level compared to the same period of 2020, with an increase in the Center (+31.2%) and in the south (+6.1%), compared to a reduction observed in the northern regions (−8.3%). When evaluating consumption in relation to hospitalisation days, an increase was observed (+36.7%), with the greatest variations in the North (+56.0%), compared to the Center (+23.7%) and to the South (+21.6%).

### 2.4. Correlation Analyses 

The analysis which investigated the relation between the number of hospitalised patients (with symptoms and intensive care assistance) and the consumption of antibiotics in 2020 (overall and by class) found a positive correlation for all classes, with the exception of carbapenems (Appendix A). The strongest correlation was found for the overall antibiotics category (*r di Pearson* hospitalisation with COVID-19 symptoms: 0.82; *r di Pearson* intensive care: 0.87) and for third generation cephalosporins (*r di Pearson* hospitalisations with COVID-19 symptoms: 0.88; *r di Pearson* intensive care: 0.91). 

## 3. Discussion

This descriptive study is the largest analysis performed in Italy of antibiotic consumption during the COVID-19 pandemic. The related data strongly suggest the need for several and urgent actions in antimicrobial stewardship strategies. After the SARS-CoV-2 pandemic, antimicrobial resistance remains one of the biggest issues in public health, both globally and in EU/EEA countries. 

According to our study results, a wide range of antibiotic prescribing in hospitalised patients was found during 2020. The total number of hospitalised patients decreased in 2020 due to elective surgery postponement or hospital admissions reduction for non-urgent conditions, in particular, during the first wave of the pandemic. However, the indicator “DDD per 1000 population per day” does not take into account the impact of the pandemic on hospitalisation, complicating the comparison of hospital antibiotic consumption with previous years and between European countries. For this reason, our study reported consumption data also using the indicator “DDD per 100 hospitalisation days”. The apparently stable antibiotic consumption in hospital settings, expressed as DDD per 1000 population per day, actually becomes a marked increase, if expressed as DDD per 100 hospitalisation days. Considering different geographic areas, the Center and South showed a decrease in terms of DDD per 1000 population per day (−8.1% and −4.5%), and an increase in terms of DDD per 100 hospitalisation days (+4.3% and +19.9%). 

The different geographical trends were likely to depend on the greater impact of the pandemic in the north in the first semester of 2020, and suggested a frequent use of antibiotics in hospitalised cases of COVID-19 in a time period where no other treatment approach seemed to be available. During the first months of the pandemic, some antibiotics had been suggested to have a potential effect against COVID-19 infection, albeit with insufficient evidence [23]. Indeed, a clearly doubled use of antibiotics in the pandemic’s first phase in some areas of northern Italy [24] has been previously already reported. One of these was azithromycin, hypothesized to have potential efficacy when combined with hydroxychloroquine [25]. This finding was observed in adults with underlying comorbidities, despite the lack of real evidence of associated bacterial infections. 

The evidence for such wide differences in non-appropriate use of antibiotics during the pandemic suggests that an unexpected infectious disease emergency has as a first consequence a relevant impact on antibiotic use. A positive correlation between the number of hospitalised patients (with symptoms and intensive care assistance) and the consumption of antibiotics in 2020 (overall and by class) found for all classes, with the exception of carbapenems. These results propose the urgent need for global clinical guidelines on how to approach infection management, not only during ordinary healthcare, but also in emergency periods. Indeed, the therapeutic approach in clinical practice might increase immediate and long-term risks of adverse events in patients and susceptibility to secondary infections, as well as aggravating antimicrobial resistance. 

The prudent and correct use of antimicrobials is a key priority for an effective response to the emergence and spread of antimicrobial resistance, requiring concerted efforts at national and regional levels, according to different local epidemiology and with close international cooperation. Based on what we observed during the SARS-CoV-2 emergency scenario, an antibiotic prescription strategy far from scientific evidence has been widely diffused. Bacterial co-infection in COVID-19 patients appeared lower than 15% [26]; therefore, it would be prudent to reserve antibiotics for patients with suspected or severe COVID-19 manifestations possibly related to comorbidities like diabetes or other predisposing clinical comorbidities. Considering the ongoing pandemic and what it has determined in terms of antibiotic use, this trend should be modified as soon as possible. Particularly, the increase of carbapenems represents a serious challenge, given the impact of the use of these antibiotics on the further development and spread of resistance. This finding could be partly explained by the tendency to use these drugs for the treatment of hospital infections caused by multidrug-resistant microorganisms [24,27]. Furthermore, especially during the second wave of the pandemic, in which the stay in hospital was longer due to improvements in patient management in hospital settings, such as for life support, patients with COVID-19 were exposed to an increased risk of hospital infections, including those caused by *Enterobacterales* that produce extended spectrum beta-lactamases (ESBLs), which could have caused an increase in the use, for invasive infections, of antibiotics with better bactericidal activity, such as carbapenems. This hypothesis, especially at the level of empirical therapy, has been recently documented by a recent review of observational studies [28]. 

One of the main reasons for the spreading of AMR is the host microbiota disruption secondary to the extensive and inappropriate use of antibiotics [29]. In addition, studies over the past 15 years have demonstrated that unique microbial communities reside on the mucosal surfaces of the gastrointestinal tract and the respiratory tract, which have both direct and indirect effects on host defense against viral infections [30]. AMR is an emergency problem involving world public health, which after the COVID-19 pandemic, will suffer even more from the improper use of antibiotics during hospital admission of patients being positive for SARS-CoV-2 infection [30,31,32,33]. 

It is very important to monitor the trend of antibiotic consumption in the context of a health emergency, in order to assess the impact of the pandemic on bacterial resistance, as well as public health in general. This appears to be a key issue since inappropriate use could have consequences in the forthcoming future, after the pandemic phase. According to research conducted by the WHO, there is in fact the risk of an accelerated spread of resistance to antimicrobials due to the excessive use of antibiotics in the course of the SARS-CoV-2 pandemic [34]. 

Our paper has the merit of being based on a national level and reporting large amounts of data about antibiotic consumption based on hospital admission/discharges and hospital days, representing not a single or multicenter experience, but national evidence-based findings. However, despite this significant amount of consumption findings, we still lack a correlation with microbiological data and related diagnosis for admitted patients in hospital setting, which would have provided a direct evidence of the correlation of consumption data to COVID-19. In fact, our report is related only to the national use of antibiotics from different national administrative data flows and non-related to diagnosis, since currently, these two flows are not yet joined in a unique database. 

This study referred to data on medicines purchased by public healthcare facilities, which could not represent the true medication intake, although they can provide an adequate consumption estimate since they refer to all purchased antibiotics in Italy, and allow a comparison between geographical areas and temporal trends evaluation. 

Nonetheless, it gives, for the first time to our knowledge, national data on drug purchase records collected through administrative databases during the COVID-19 pandemic, showing how antibiotics are still not correctly used in hospital settings, particularly during an emergency period. 

## 4. Materials and Methods

For the analysis of hospital consumption, data on antibiotics purchased by all Italian public healthcare facilities and only dispensed within hospitals collected through the information flow called “traceability of medicines” were considered. To estimate the consumption in the outpatient setting, the “OsMed database”, collecting data regarding medicines dispensed by community pharmacies, was also used. Data on the number of hospitalisation days were extracted from the “hospital discharge form” flow provided by the Italian Minister of Health [35].

Since administrative databases were used, all data used for this National Report were related to the use of antibiotics in Italy and therefore non-related to diagnosis per admitted patients in hospital setting.

Consumption data were arranged according to the Anatomical Therapeutic Chemical (ATC) classification established by the WHO Collaborating Center (WHOCC) for Drug Statistics Methodology following: antibiotics for systemic use (J01), fluoroquinolones (J01MA), penicillins (J01CA-CE-CF), third-generation cephalosporins (J01DD), macrolides (J01FA), and carbapenems (J01DH). 

Drug consumption was measured as number of Defined Daily Dose (DDD), which is the assumed average maintenance dose per day for a drug used for its main indication in adults. It represents a standard in performing valid and reliable cross-national or longitudinal studies on drug consumption. Since DDD values of some medicines may change over time because of alterations in the main indication, or regulatory amendments for the recommended or prescribed daily dose, all historical data were retrospectively adjusted to the latest version of the DDD/ATC index. The indicators calculated as “number of DDD per 1000 inhabitants per day” and “number of DDD per 100 hospitalisation days” were used. 

The 19 Italian Regions and the two Autonomous Provinces were grouped into three geographic areas, northern, center, and southern, according to the Italian National Statistics Institute’s classification [36,37].

Finally, a correlation analysis between the number of patients hospitalised for COVID-19 (with symptoms and need of intensive care) and the number of doses of antibiotics purchased by public healthcare facilities in the year 2020 was performed; data about the number of COVID-19 hospitalisations were extracted from the interactive dashboard made available by the Italian Department of Civil Protection. This analysis was carried out considering the overall consumption of antibiotics and the consumption of four specific antibiotic classes: third-generation cephalosporins, carbapenems, macrolides, and fluoroquinolones. The Pearson correlation coefficient (r) was used for measuring the linear correlation. The test for significance is not provided, since the data collected through administrative databases referred to all Italian public health facilities.

## 5. Conclusions

In conclusion, during the COVID-19 pandemic, a substantial overall increase in hospital antibiotic consumption was registered at national level in Italy, even correlated to noteworthy geographical differences. Our results strongly underline some important findings. First, despite the available international and national guidelines, antibiotics are still used in Italy according to different strategies with a variation in terms of geographical distribution. The main reason for this trend could be found in the dichotomy between perceived and real in terms of guidelines, expert panel, or consensus, that could be related to differences in local experiences and epidemiology that may not match with other proposed and theoretical models not fully applicable in clinical settings, particularly in emergency. Therefore, a novel approach in antibiotic use and antimicrobial stewardship should be adopted. Particularly, it could be based on considering local epidemiology coupled to national and international evidence-based guidelines.

Secondly, the COVID-19 pandemic showed that, in the case of an emergency scenario, antibiotics are the first-used class of medications. This evidence strongly suggests the weight of the perceived–real dichotomy in healthcare settings. Antibiotic consumption data represent a fundamental starting point but it requires integration with microbiological data to have a wider and more accurate assessment of AMR and inappropriate use of antibiotics. Particularly, it should also be based on age classification to better address future strategy. Therefore, further studies on national data will be performed integrating more stakeholders of the Italian national healthcare system, including clinical information reported in Italian hospital discharge forms and microbiological findings. 

## Figures and Tables

**Figure 1 antibiotics-12-00168-f001:**
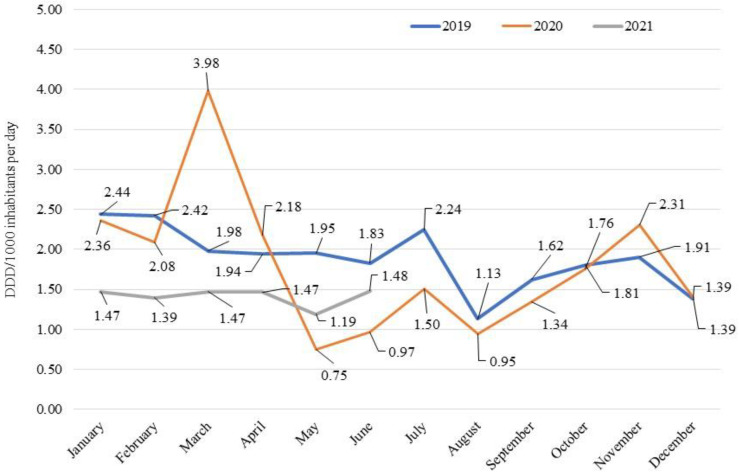
Consumption (DDD/1000 inhabitants per day) on a monthly basis of antibiotics for systemic use (J01) in the hospital setting: 2019–2021 comparison.

**Figure 2 antibiotics-12-00168-f002:**
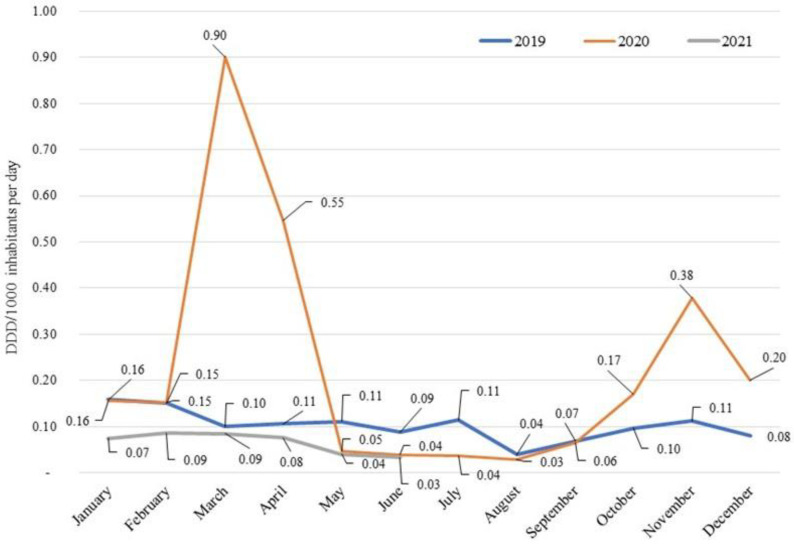
Consumption (DDD/1000 inhabitants per day) on a monthly basis of azithromycin in the hospital setting: 2019–2021 comparison.

**Table 1 antibiotics-12-00168-t001:** Consumption of antibiotics for systemic use (J01) reimbursed by Italian NHCS (community and hospital setting) and antibiotics prescribed exclusively in hospital settings during 2020.

	Italy	Northern	Center	Southern
Reimbursed by NHCS (community and hospital setting)				
DDD/1000 inhabitants per day	13.8	11.4	14.2	17.0
∆% 2020–2019	−21.7	−21.6	−24.1	−20.5
Hospital setting				
DDD/1000 inhabitants per day	1.7	2.1	1.6	1.3
∆% 2020–2019	0.8	7.3	−8.1	−4.5
DDD/100 hospitalisation days	92.1	94.9	90.8	87.7
∆% 2020–2019	19.3	24.7	4.3	19.9

**Table 2 antibiotics-12-00168-t002:** Consumption (DDD/1000 inhabitants per day) of antibiotics for systemic use (J01) in the hospital setting: comparison 2019–2021.

Geographical Areas	I Sem 2019	I Sem 2020	I Sem 2021	Δ% 20–19	Δ% 21–20	II Sem 2019	II Sem 2020	Δ% 20–19
**Italy**	**2.1**	**2.1**	**1.4**	**−1.6**	**−31.4**	**1.7**	**1.5**	**−8.5**
North	2.4	2.6	1.6	5.5	−38.0	2.0	1.8	−9.6
Center	2.1	1.9	1.4	−10.5	−23.6	1.6	1.4	−9.4
South	1.6	1.5	1.2	−9.7	−20.6	1.3	1.3	−5.6

**Table 3 antibiotics-12-00168-t003:** Consumption (DDD/1000 inhabitants per day) of azithromycin, third-generation cephalosporins, and carbapenems in the hospital setting: comparison 2019–2021.

Geographical Areas	I Sem 2019	I Sem 2020	I Sem 2021	Δ% 20–19	Δ% 21–20	II Sem 2019	II Sem 2020	Δ% 20–19
***Azithromycin* (*J01FA10*)**								
**Italy**	**0.1**	**0.3**	**0.1**	**159.9**	**−78.7**	**0.1**	**0.1**	**71.5**
North	0.2	0.5	0.1	192.0	−83.7	0.1	0.1	30.2
Center	0.1	0.2	0.1	69.1	−67.7	0.1	0.1	57.4
South	0.1	0.1	0.0	145.6	−63.7	0.0	0.2	221.2
***Third-generation cephalosporins* (*J01DD*)**								
**Italy**	**0.4**	**0.4**	**0.3**	**0.5**	**−31.0**	**0.3**	**0.3**	**0.6**
North	0.4	0.4	0.3	20.9	−42.9	0.3	0.3	5.0
Center	0.4	0.3	0.3	−13.3	−19.6	0.3	0.3	−4.2
South	0.3	0.3	0.2	−21.8	−11.0	0.3	0.2	−3.1
***Carbapenems* (*J01DH*)**								
**Italy**	**0.05**	**0.06**	**0.06**	**32.8**	**−8.3**	**0.05**	**0.07**	**33.5**
North	0.04	0.05	0.06	19.0	31.2	0.04	0.04	0.0
Center	0.05	0.06	0.07	25.3	6.1	0.06	0.05	−25.5
South	0.05	0.06	0.06	27.8	3.1	0.05	0.06	5.3

## Data Availability

Data were analysed under licence and are not available for public sharing.

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
