# Peer review of "Hospital Antibiotic Use during COVID-19 Pandemic in Italy"

_antibiotics, 2023, doi:10.3390/antibiotics12010168_

Round 1

Reviewer 1 Report

The manuscript from Perrella and colleagues presented a study investigating antibiotic use in hospital setting in Italy affected by COVID-19 pandemics. They use the consumption data on reimbursed antibiotics dispensed in hospital setting between 2019 and 2021. They reported a substantial overall increase in antibiotic prescriptions was observed in Italy during COVID-19 pandemic, with a variance in terms of geographical distribution and prescription strategies. They also found that, COVID-19 pandemic influenced the use of selected class of antibiotics. These findings suggested that new approaches or strategies to antimicrobial stewardship should be proposed. However, I found the evidence are circumstantial and the manuscript could be improved.

Major comments

1.      The authors compared antibiotic consumption data between 2019 and 2020 as the impact from COVID-19. To some extent, it could not accurately reflect the real “mis-use” of antibiotics cause by COVID-19. It would be more convincing if the authors could use the consumption data from non-bacterial infected patients.

2.      The authors found geographical trends were different among the areas compared and suggested “that this trend was likely to depend on the greater impact of the pandemic and suggest a frequent use of antibiotics in hospitalized cases of COVID-19 in a time period where no other treatment approach seemed to be available.” There is no evidence in the manuscript could support this. It would be better if the authors could show some evidence of greater impact of the pandemic in the North in the first semester of 2020.

3.      Although the authors observed some variations among these compared areas, is it possible that at the year of 2020 a bacterial outbreak happened? Again, the consumption data of non-bacterial infected could be more useful. Or an investigation of COVID-19 infections in these areas could serve as circumstantial evidence.

Minor comments:

1.      Please make sure the acronyms were placed at the first place it showed up. For example: Defined Daily Dose (DDD)

2.      The abstract should be concise.

Author Response

Major comments

  1. The authors compared antibiotic consumption data between 2019 and 2020 as the impact from COVID-19. To some extent, it could not accurately reflect the real “mis-use” of antibiotics cause by COVID-19. It would be more convincing if the authors could use the consumption data from non-bacterial infected patients.

Dear Referee thank you for your suggestion on the use of consumption data from non-bacterial infected patients. Unfortunately, since our national report is on use of antibiotics and non-related to diagnosis for admitted patients in hospital setting we cannot have access to this kind of data. In fact, these are data from different national administrative data flows. Neverthless, we have better specified the origin of our data in the methods and discussion sections..

  1. The authors found geographical trends were different among the areas compared and suggested “that this trend was likely to depend on the greater impact of the pandemic and suggest a frequent use of antibiotics in hospitalized cases of COVID-19 in a time period where no other treatment approach seemed to be available.” There is no evidence in the manuscript could support this. It would be better if the authors could show some evidence of greater impact of the pandemic in the North in the first semester of 2020.

Thank you for your comment. We have improved the introduction and discussion sections adding some evidence of the impact of the pandemic in Northern Italy.

  1. Although the authors observed some variations among these compared areas, is it possible that at the year of 2020 a bacterial outbreak happened? Again, the consumption data of non-bacterial infected could be more useful. Or an investigation of COVID-19 infections in these areas could serve as circumstantial evidence.

Dear referee, the observation you have done should be the reason of our decision to conduct a microbiological study matching the microbiological data with antimicrobials consumption in the future. Therefore, as we have previuolsly undelined we have no administrative consumption data matching the microbiological findings.

Minor comments:

  1. Please make sure the acronyms were placed at the first place it showed up. For example: Defined Daily Dose (DDD)
  2. The abstract should be concise.

We have checked the acronyms across the manuscript and shortened the abstract in line with your suggestion.  

Reviewer 2 Report

Suggestions included in attached file

Author Response

General Comments: the manuscript contains information regarding antimicrobial stewardship. It has been presented very well. Information’s related to antimicrobial consumption, surveillance and monitoring is currently the most concerning issue to prevent any antimicrobial resistance and emergence of superbugs. Hence, manuscript will gather attention of wide range of readers across the globe. Order of manuscript sections needs rearrangement. In conclusion, the manuscript can be accepted for publication.

Dear Referee, we have rearranged the order of manuscript according to journal author’s instructions. Thank you very much for your note and comment.

Reviewer 3 Report

Here are a few comments that may help improve the paper:

1.     Consider providing more context and background information in the introduction. The current introduction briefly mentions antimicrobial resistance (AMR) and the impact of the COVID-19 pandemic on antibiotic use, but it could benefit from a more thorough discussion of these issues and their importance.

2.     Clarify the research question and objectives of the study. It's not clear from the introduction what the specific aim of the study is, or how the authors plan to evaluate the impact of the COVID-19 pandemic on antibiotic consumption in hospitals.

3.     Provide more detail about the methods used in the study. The manuscript mentions that consumption data on reimbursed antibiotics and data on hospitalizations days were collected, but it's not clear how these data were obtained or what specific sources were used.

4.     Consider providing more information about the statistical analysis performed. The manuscript mentions that a descriptive analysis and a parametric statistical assessment with Pearson correlation were performed, but it's not clear what specific statistical tests were used or how the results of these tests were interpreted.

5.     Add more discussion of the results and their implications. The manuscript presents some data on antibiotic consumption in hospitals during the COVID-19 pandemic, but it could benefit from a more thorough analysis and interpretation of these data.

6.     Consider including a conclusion section that summarizes the main findings of the study and discusses their implications for future research and practice.

7.     The numerical results should be accompanied by appropriate statistical analysis results, if possible.

8.     All over the manuscript, several grammatical and syntax errors were detected. Kindly polish your manuscript.

 I hope these suggestions will be helpful. Good luck with your paper!

Author Response

Here are a few comments that may help improve the paper:

  1. Consider providing more context and background information in the introduction. The current introduction briefly mentions antimicrobial resistance (AMR) and the impact of the COVID-19 pandemic on antibiotic use, but it could benefit from a more thorough discussion of these issues and their importance.

Dear referee we are very happy about your comment that gives us the opportunity to improve the introduction and the discussion as you suggested.  

  1. Clarify the research question and objectives of the study. It's not clear from the introduction what the specific aim of the study is, or how the authors plan to evaluate the impact of the COVID-19 pandemic on antibiotic consumption in hospitals.

As for the first comment you reported for our paper, to better clarify we have provided the aim and the evaluation strategy about the impact of the COVID-19 pandemic on antibiotic consumption in hospitals.

  1. Provide more detail about the methods used in the study. The manuscript mentions that consumption data on reimbursed antibiotics and data on hospitalizations days were collected, but it's not clear how these data were obtained or what specific sources were used.

We have clarified what you have requested.

  1. Consider providing more information about the statistical analysis performed. The manuscript mentions that a descriptive analysis and a parametric statistical assessment with Pearson correlation were performed, but it's not clear what specific statistical tests were used or how the results of these tests were interpreted.

The test for significance is not provided since the data collected through administrative database, are referred to the whole Italian population.

  1. Add more discussion of the results and their implications. The manuscript presents some data on antibiotic consumption in hospitals during the COVID-19 pandemic, but it could benefit from a more thorough analysis and interpretation of these data.

Dear referee thank you for your suggestion, we have improved the paper according to your proposal adding more data and related intepretation for the antibiotic consumption data in hospitals during the COVID-19 pandemic.

  1. Consider including a conclusion section that summarizes the main findings of the study and discusses their implications for future research and practice.

Dear referee we have attached to your comment the conclusion section that summarizes the main findings of the study and discusses their implications for future research and practice. You will find it in the final version of the paper.

“Our results strongly underline some important findings. First, despite the available international and national guidelines recommendations, antibiotics are still used according to different strategies with a significant variation in terms of geographical distribution and prescription strategies. The main reason of this trend could be found in the dichotomy between perceived-real significance of guidelines, expert panel or consensus. This could be related to differences in local experiences and epidemiology that may not be compatible with other proposed and theorical models. Therefore, a novel approach in the antibiotic use and antimicrobial stewardship should be adopted. Particularly it could be based on strategies involving a multidisciplinary approach including local evidences compared with national and international evidence-based guidelines.

Secondly, the COVID-19 pandemic showed that, in case of an emergency scenario, antibiotics are the first class of medications, once the patient is admitted to the hospital. This basically bears the weight of the perceived-real dichotomy in healthcare settings. Antibiotics consumption data represent a fundamental starting point but they require to be integrated with microbiological data to have a wider and more accurate assessment of AMR and inappropriate use of antibiotics. Particularly, it should also be based on age classification to better address future strategy. Therefore, further studies on national data will be performed integrating more stakeholders of Italian national healthcare system including clinical information reported in the Italian hospital discharge forms and microbiological findings.”

  1. The numerical results should be accompanied by appropriate statistical analysis results, if possible.

The test for significance is not provided since data collected through administrative database are referred to the whole Italian population. However we have tried to improve at our best the materials and methods section.

  1. All over the manuscript, several grammatical and syntax errors were detected. Kindly polish your manuscript.

Dear referee we are sorry about grammatical and syntax errors, we have checked and corrected them in the whole paper.

Round 2

Reviewer 1 Report

The abstract can be futher improved.

Author Response

Dear referee,

we have improved the abstract, as you suggestes. 

Thank you.